# Late-Night Overeating or Low-Quality Food Choices Late at Night Are Associated with Subclinical Vascular Damage in Patients at Increased Cardiovascular Risk

**DOI:** 10.3390/nu14030470

**Published:** 2022-01-21

**Authors:** Eirini D. Basdeki, Konstantina Koumi, Christiana Tsirimiagkou, Antonios Argyris, Stavri Chrysostomou, Petros P. Sfikakis, Athanase D. Protogerou, Kalliopi Karatzi

**Affiliations:** 1Cardiovascular Prevention & Research Unit, Clinic & Laboratory of Pathophysiology, Department of Medicine, National and Kapodistrian University of Athens, 15772 Athens, Greece; eirinibasdeki@gmail.com (E.D.B.); tsirimiagou.ch@gmail.com (C.T.); and1dr@gmail.com (A.A.); aprotog@med.uoa.gr (A.D.P.); 2Department of Nutrition and Dietetics, School of Health Science and Education, Harokopio University of Athens, 17671 Kallithea, Greece; 3Department of Life Sciences, School of Sciences, European University of Cyprus, Nicosia 2404, Cyprus; ck192255@students.euc.ac.cy (K.K.); s.chrysostomou@euc.ac.cy (S.C.); 41st Department of Propaedeutic Internal Medicine, Medical School, National and Kapodistrian University of Athens, 15772 Athens, Greece; psfikakis@med.uoa.gr; 5Laboratory of Dietetics and Quality of Life, Department of Food Science & Human Nutrition, Agricultural University of Athens, 11855 Athens, Greece

**Keywords:** late-night overeating, subclinical vascular damage, cardiovascular disease, AIx, PWV (pulse wave velocity), carotid plaques

## Abstract

Late-night overeating (LNO) is associated with several cardiovascular disease (CVD) risk factors. Limited data exist regarding the association between late-night (LN) systematic food consumption, LNO, and LN poor food quality with subclinical vascular damage (SVD) which precedes the onset of CVD. This study aimed to investigate the above associations with SVD in a large sample of adults, free of established CVD, with one or more CVD risk factors. In total, 901 adults (45.2% males) underwent anthropometric, dietary (through two 24 h dietary recalls) and vascular assessment. LN systematic eating was defined as consumption of food after 19:00 h in both dietary recalls and LNO was defined as systematic consumption of >40% of daily total energy intake (dTEI) after 19:00 h. Systematic LN food consumption was inversely associated with diastolic blood pressure (DBP) (−1.44 95% C.I. (−2.76, −0.12)) after adjusting for age, sex, hypertension, diabetes, dyslipidemia, smoking, BMI and dTEI. LNO was positively associated with existence of carotid plaques (1.70 95% C.I. (1.07, 2.68)), while LN increased consumption of red meat, refined grains and wine and low consumption of whole wheat grains was positively associated with Aix (Augmentation Index) (0.84 95% C.I. (0.09, 1.59)), after adjusting for all the mentioned confounders. Systematic LN eating is associated with lower DBP while systematic LNO and consumption of poor-quality food late at night, is associated with SVD. Further research is needed to define more accurately the impact of LN eating habits on vascular health.

## 1. Introduction

Late-night overeating is considered the consumption of a large amount of daily total energy intake (dTEI) late in the day and/or during the night. However, so far there is no standardized and widely acceptable definition for late-night overeating. Different approaches have been suggested in relevant studies, such as intake of >25% of dTEI between 19:00 to 4:59 h or ≥33% at night, an occasion of eating in between sleep, consumption of dinner within two hours before bedtime, and intake of >35% of dTEI after 20:00 h [1,2,3,4]. Modern lifestyles induce late-night overeating, which has been associated with generally increased energy intake within the day and with low food quality [1,2,3,5]. There are many groups of people that follow (or are characterized by) this type of eating behavior either due to lifestyle, i.e., shift-workers, or in the presence of night eating syndrome (NES), or due to religious particularities (Muslims in the period of Ramadan) [1,2,6,7,8].

Subclinical vascular damage (SVD) precedes the onset of CVD; therefore, its detection can contribute to identifying potential risk factors and improving timely prevention strategies [9]. Vascular damage is either functional or structural and can be examined by a wide variety of non-invasive vascular indices [9,10]. Blood pressure (BP) (systolic and diastolic (SBP, DBP)), augmentation index (AIx) (pressure wave reflections), pulse wave velocity (PWV) (arterial stiffening), carotid intima media thickness (IMT) (arterial remodeling) and carotid and femoral plaques (atheromatosis) are major indices of arterial dysfunction and damage [9,10,11].

Late-night overeating has been associated with several CVD risk factors, such as obesity, hypertension, glycemic and lipid profile disorders and increased CRP [2,3,4,5,12,13,14]. Several characteristics of this eating behavior may contribute to these associations. For example, increased daily energy intake, which is reported in late-night overeating, paralleled with reduction of food thermogenesis and resting metabolic rate can lead to positive energy balance and obesity [2,3,5]. Moreover, there are many studies available that support the association between obesity or hypertension and SVD [15,16,17,18,19]. Regarding the quality of diet of late-night overeaters, data also indicate an indirect association. Ferreira et al. found that dietary choices after dinner are usually salty snacks [16], while salt intake has already been associated with arterial stiffness and endothelial dysfunction [20,21,22]. Thus, all the aforementioned data, indirectly suggest a potential association between late-night overeating and SVD; however, only one study has directly investigated this association [23]. Our hypothesis is that presence of late-night overeating is associated with SVD indices. The aim of this study is to investigate this potential association in a large sample of adults with one or more CVD risk factors. To test this hypothesis, we will take into consideration both the quantity and quality of food in late-night eating.

## 2. Materials and Methods

### 2.1. The Study

The present study has a cross-sectional design, and it was conducted at the Laiko University Hospital, Athens, Greece. The study is in accordance with the Declaration of Helsinki and its later amendments, it was approved by the Bioethics Committee of the University Hospital and all participants signed a written consent form before entering the study.

### 2.2. Study Population

The study sample consisted of 901 adults (45.2% males) with one or more CVD risk factors (suspected or established, treated or untreated hypertension, diabetes mellitus, dyslipidemia and/or chronic inflammatory diseases) but free of established CVD. Individuals with cancer, liver disease, gastrointestinal health issues or eating disorders were excluded from the study. All patients abstained from food, drink, or any medication for 12 h prior to their visit in the laboratory for the conduction of anthropometric measurements and vascular assessment.

### 2.3. Definition of CVD Risk Factors

Hypertension was defined by the use of antihypertensive drugs and/or office blood pressure measurement higher than 139/89 (average of 3 sequential readings with 1 min interval in the supine position after at least 10 min of rest (Microlife WatchBP Office, Microlife AG, Widnau, Switzerland). Dyslipidemia was defined on the basis of treatment with lipid modifying drugs or low-density lipoprotein cholesterol level > 160 mg/dL. Current smoking was defined by the use of at least 1 cigarette per day each day of the week; ex smoking was defined as discontinuation for more than 6 months. Body mass index was calculated as weight/(height^2^) (Kg/m^2^) and used as a marker of obesity. Family history of premature CVD was defined as the presence of coronary heart disease in a 1st degree relative under the age of 55 years for males and 65 years for females. 

### 2.4. Anthropometric Measurements

Participants’ body weight (by weight scale Tanita Body Composition Analyzer, BC−418), height (by stadiometer SECA 213) and BMI were measured.

### 2.5. Vascular Assessment

The study’s participants underwent vascular assessment, performed by the same examiner. BP measurement was conducted three sequential times, after a 10-min rest of the volunteers in the supine position. The average of the three measurements was used as the final BP value. Central pressures, pressure wave reflections by AIx and carotid to femoral PWV were measured by tonometry methods in order to assess aortic stiffness, as described previously [15]. Subclinical atheromatosis was assessed by high-resolution ultrasound. IMT was calculated in right and left carotid artery, in three different spots: common carotid artery, carotid bulb and internal carotid artery, as well as at the femoral artery. The mean IMT of the three measurements of the common carotid artery was used. Atheromatic plaques were defined as a local increase of the IMT of >50% compared to the surrounding vessel wall, an IMT > 1.5 mm or local thickening > 0.5 mm [24].

### 2.6. Dietary Intake Assessment and Definition of Late-Night Overeating

Following vascular assessment, two 24 h dietary recalls were conducted (one weekday and one weekend day) with an interval of at least one week in between, by trained dieticians. Data were analyzed in food groups, where separation was based on food equivalents with further separation in subgroups where needed. Analysis of energy content, macronutrients and micronutrients was also conducted by using the Nutritionist pro Software (Axxya Systems Nutritionist Pro TM 2011, Athens, Greece). Regarding late-night overeating, food consumption data for the evening hours were recorded separately. Energy intake, macro- and micronutrients consumed after 19:00 h and their percentage in dTEI were calculated. Systematic late-night food consumption was defined as food intake after 19:00 h in both dietary recalls and non-systematic late-night food consumption was defined as food intake after 19:00 h in 0 or 1 dietary recall. Systematic late-night food consumption was further divided into systematic late-night eating (consumption of <40% of dTEI after 19:00 h) and systematic late-night overeating (consumption of >40% of dTEI after 19:00). Quality of dinner among systematic late-night food consumers was performed using 3 dietary patterns, which statistically emerged a posteriori (description below) based on the food consumed after 19:00 h.

### 2.7. Statistical Analysis

Statistical analyses were conducted using statistical package 21.0 (SPSS: Statistical Package for Social Sciences, SPSS Inc., Chicago, IL, USA). Level of statistical significance was set at *p* ≤ 0.05 for all analyses.

#### 2.7.1. Analyses Based on Food Quantity after 19:00 h

The normality of distribution of variables was checked using the Kolmogorov–Smirnov test. Categorical variables are presented as relative frequencies (%) and continuous as mean ± standard deviation (SD). Paired *t*-test analysis was applied to address differences between non-systematic late-night food consumption and systematic late-night food consumption, and between systematic late-night eating and systematic late-night overeating. Multiple linear and logistic regression analysis was performed to determine the independent associations between vascular biomarkers and (i) systematic late-night food consumption (vs. non-systematic), and (ii) systematic late-night overeating (vs. systematic late-night eating). Results are presented as standardized beta coefficients and 95% confidence intervals (CI) (for peripheral and central SBP, DBP, PWV, AI75, right and left cIMT) or as odds ratio (OR) expressed as ExpB and 95% CIs (for existence of total plaques, carotid plaques and femoral plaques). Three different models were applied to assess the above associations: Model 1: adjusted for age and sex;Model 2: adjusted for age, sex, hypertension, diabetes mellitus, dyslipidemia, smoking and BMI;Model 3: adjusted for age, sex, hypertension, diabetes mellitus, dyslipidemia, smoking, BMI and dTEI.

#### 2.7.2. Analyses Based on Food Quality after 19:00 h

Principal Components Analysis (PSA) was applied to identify dietary patterns (DP) among study participants with systematic late-night food consumption. The Kaiser–Meyer–Olkin (KMO) criterion was applied, and it was equal to 0.592. The orthogonal rotation (varimax option) was used to derive optimal non-correlated components (DPs) and the Bartlett’s method was used to estimate factor scores. The selection of the optimal number of components was based on an eigen value > 1 (Kaizer criterion) and it was further corroborated by visual assessment of the scree plot, retaining only components on the steep slope. If one of the initial variables correlated with more than one component, it participated in the interpretation of the DP in which it displayed the highest coefficient value. Three DPs emerged after PCA. Multiple linear and logistic regression analysis were applied to determine possible associations between DPs and subclinical vascular biomarkers. Results are presented as standardized beta coefficients and 95% confidence intervals (CI) (for peripheral and central SBP, DBP, PWV, AI75, right and left cIMT) or as odds ratio (OR) expressed as ExpB and 95% CIs (for the existence of total plaques, carotid plaques, and femoral plaques). The aforementioned models that were applied in food quantity analyses, were also applied in this analysis, in order to assess the above associations.

## 3. Results

The study sample consisted of 901 CVD free adults (45.2% males), with mean age 52.4 ± 13.81 years, mean BMI 28.0 ± 5.53 Kg/m^2^ with at least one risk factor for CVD. One hundred and seventy-eight participants were non-systematic late-night food consumers and 723 were systematic late-night food consumers. Systematic late-night food consumers were divided into two subgroups: 572 systematic late-night eaters (<40% of dTEI) and 151 systematic late-night overeaters (>40% of dTEI). After an unadjusted *T*-test comparison, systematic late-night food consumers present with significantly better values of several vascular biomarkers (SBP, DBP, central SBP, existence of total and femoral plaques) compared to non-systematic late-night food consumers. Systematic late-night food consumers of <40% of dTEI had significantly better values of HDL and LDL cholesterol, but higher values of PWV and AIx compared to late-night overeaters. Descriptive and clinical characteristics of study’s participants are presented in Table 1.

Dietary intake of the study participants is presented in Table 2. Results emerged after conduction of unadjusted *t*-test comparisons. No statistically significant differences were observed between dTEI (*p* = 0.172) and macronutrients intake expressed as percentage of dTEI (*p* = 0.490 for carbohydrates, *p* = 0.058 for protein and *p* = 0.174 for fat) between systematic vs. non-systematic late-night food consumers. However, in systematic late-night food consumers lower consumption of legumes (*p* = 0.006) and higher consumption of cheese (low (*p* = 0.012) and full fat (*p* = 0.027)) was observed. On the other hand, systematic late-night overeaters had significantly higher dTEI compared to systematic late-night food consumers of <40% of dTEI (1886.45 ± 740.84 vs. 1644.39 ± 558.02, *p* = 0.000 respectively), but lower percentage of carbohydrates (40.91 ± 10.77 vs. 42.71 ± 9.36, *p* = 0.043). Moreover, systematic late-night overeaters tended to have worst diet quality compared with systematic late-night food consumers of <40% of dTEI, consuming less low-fat dairy products (0.20 ± 0.42 vs. 0.25 ± 0.40, *p* = 0.001), fruits and fresh juices and more refined grains (3.66 ± 2.57 vs. 2.80 ± 2.26, *p* = 0.000), dressings and soft drinks with sugar (0.11 ± 0.32 vs. 0.06 ± 0.23, *p* = 0.026).

The associations of subclinical vascular biomarkers with systematic food consumption late at night compared to non-systematic food consumption late at night, and with systematic late-night eating compared to systematic late-night overeating, are presented in Table 3. Systematic vs. non-systematic consumption of food late at night was negatively associated with DBP (−1.45 (−2.76, −0.13)) after adjusting for age, sex, hypertension, diabetes mellitus, dyslipidemia, smoking and BMI, which remained significant even after adjustment for dTEI (−1.44 (−2.76, −0.12)). Furthermore, systematic late-night overeating (compared to systematic late-night food consumption of <40% of dTEI) was positively associated with existence of carotid plaques (1.61 (1.02, 2.53)) after adjusting for age, sex, hypertension, diabetes mellitus, dyslipidemia, smoking and BMI, which remained significant after adjustment for dTEI (1.70 (1.07, 2.68)). The performed analysis was repeated once again adjusting for all the already mentioned confounders and inserting in the model as additional confounders drugs for diabetes, dyslipidemia and/or hypertension, which resulted in no alterations in the reported results.

Table 4 summarizes the loadings of the factors retained from the PCA and identified three DPs, explaining 41.57% of systematic food consumers late at night. The derived DPs were as follows: high consumption of olive oil, vegetables and fish (DP1); high consumption of cold cuts, full fat cheese, sweets with sugar and saturated fat and low consumption of fruits and fresh juices (DP2); and high consumption of red meat, wine, refined grains and low consumption of whole wheat grains (DP3).

Finally, Table 5 presents the results of multiple linear and logistic regression analysis between the three DPs of systematic consumers of food late at night and subclinical vascular biomarkers. DP3 had marginally not statistically significant association with AIx (0.63 (−0.11, 1.37), *p* = 0.094) after adjusting for age, sex, hypertension, diabetes mellitus, dyslipidemia, smoking and BMI. However, after adjustment for dTEI, DP3 was positively associated with AIx (0.84 (0.09, 1.59), *p* = 0.028). In further analysis, after adjusting for age, sex, BMI, dyslipidemia, diabetes mellitus, hypertension, smoking and total energy intake after 19.00 (instead of dTEI), the positive association with Aix remained unaltered (B = 0.96, 95%CI (0.16, 1.77), *p* = 0.019) (data are not presented in tables).

## 4. Discussion

In the present study, we aimed to examine the potential associations of eating late at night with SVD, by taking into consideration two major parameters of eating, i.e., total energy intake (food quantity) and dietary patterns (food quality) in adults with CVD risk factors. Our findings suggest that systematic late-night overeating (consumption of >40% of dTEI late at night), non-systematic late-night food consumption and low-quality food choices late at night is associated with SVD. Regarding food quantity, a negative association between systematic late-night food consumption and DBP, and a positive association between late-night overeating with the existence of carotid plaques was observed. Furthermore, regarding food quality, there was a positive association between the dietary pattern characterized by high consumption of red meat, wine, refined grains and low consumption of whole wheat grains with AIx.

To begin with, there was no difference in dTEI between systematic or non-systematic late-night food consumption. However, dTEI was significantly higher in systematic late-night overeaters compared to systematic late-night food consumers of <40% of dTEI. Our findings are in accordance with the results of another cross-sectional study, where it was also found that late-night overeating was associated with high total caloric intake [25]. Regarding food quality, systematic vs. non-systematic late-night food consumers had lower legume and higher cheese consumption. Furthermore, systematic late-night overeaters had significantly lower consumption of low-fat dairy products, fruits and fresh juices and higher consumption of refined grains low-fat cheese, dressings and sugar sweetened soft drinks, compared to systematic late-night food consumers of <40% of dTEI. Late-night overeating and quality of food has barely been examined previously. There is only one available relevant review which also concluded that systematic food consumption of >35% dTEI after 20:00 h was associated with low fruit consumption and high consumption of grains, but further separation for the type of the grain products was not conducted [1].

Non-systematic compared to systematic late-night food consumers had higher BP levels (peripheral and central SBP and DBP); however, this might be attributed to the high percentage of hypertension in the study’s participants. In a retrospective cohort study, Kakamu et al. found that eating before bedtime could be a risk factor for hypertension in normotensive elderly [26], but the current population is not only normotensive adults, which does not allow comparisons between the findings of the two studies. On the other hand, in the present study, it was shown that consumption of a meal at the end of the day, instead of systematically skipping food consumption late at night, is associated with lower DBP levels. Although, there are no relevant data, interestingly in a cross-sectional study conducted in adolescents [27] it was shown that consumption of dinner was inversely associated with some CVD risk factors (e.g., obesity) and supported the possible aggravating effects of dinner skipping on CV health. These findings are indirectly in agreement with the findings of the present study. The group of systematic late-night overeaters had significantly lower PWV and AIx measurements compared to systematic late-night food consumers of >40% of dTEI; however, this result may be attributed to the lower age of this group, since arterial stiffness is normally increased due to aging [28]. 

Late-night overeating was positively associated with the existence of carotid plaques. There is only one available study examining late-night overeating and its association with SVD, but their analysis showed a positive association only with PWV [23]. There are no available data connecting late-night overeating with the existence of carotid plaques; however, an indirect connection can be hypothesized. Snacks after dinner tend to be energy dense, salty and are mainly combined with sedentary activities, e.g., watching television [16]. All the above behaviors are strongly associated with the existence of carotid plaques and vascular dysfunction in general [29,30]. Interestingly, in a cross-sectional study conducted in Greece, TV viewing of >21 h/week increased by 80% the odds of carotid atheromatic plaque existence, compared with viewing ≤ 7 h/week [30].

Regarding the quality of food in systematic late-night food consumers, it was found that a DP characterized by increased consumption of red meat, wine, refined grains and decreased consumption of whole grains was positively associated with AIx, regardless of various confounders, such as age, BMI, dTEI, and energy intake of any meal after 19:00 h. Since the presented association is independent of dTEI and energy intake after 19:00 h, quality of food after 19:00 h, independently of quantity is associated with AIx. Although, there is no available study examining consumption of the above DP with AIx, components of the DP have been studied separately. Red meat consumption in young students was positively associated with AIx [31]. Excessive alcohol consumption, regardless of the type of beverage, leads to an increase in AIx [15]. Type of grains might also be associated with AIx. Whole grains have a lower glycemic index (GI) and glycemic load (GL) compared to refined grains. In a cross-sectional study with 1553 participants, free of CVD, both GI and GL were positively associated with AIx [32]. Therefore, a DP comprised of the above food choices might have an aggravating effect on vascular health.

The present study is one of the few available studies to examine the associations of (i) systematic vs. non-systematic late night food consumption, (ii) systematic late-night food consumption of <40% of dTEI vs. systematic late-night overeating and (iii) the quality of food consumed late at night among systematic late-night food consumers, with different SVD biomarkers. In terms of nutritional evaluation, the 24-h recalls which were applied in our study, are based on participants’ memory, thus they are prone to recall bias. However, in our study, we use the multiple pass method during the dietary assessment, in order to minimize the potential error. Furthermore, 24-h recalls remain a valuable tool for dietary evaluation in studies, as they not only allow the assessment of dietary intake of the population, but they also provide the advantage of additional information recording, regarding the overall quality of nutrition. It is also important to note, that dietary intake analysis was conducted by Nutritionist Pro, a program that mainly contains American foods, and there may be variation in food composition from country to country. However, a variety of common greek foods and their nutritional analysis were registered into the program, to minimize possible declinations. Furthermore, regarding vascular assessment, it was conducted by the same examiner with the same equipment, thus reducing any possible measurement error. Another possible limitation of the study is that the majority of the participants were late-night eaters. However, this reflects a very typical dietary behavior in Greece as late-night overeating is very common among Greeks. Lack of data on participants’ physical activity is also a limitation of the present study. Finally, although the cross-sectional design of this study does not allow to establish a causative relationship, we performed a statistical analysis with detailed adjustment for all potential confounders, in order to limit potential bias, as possible.

## 5. Conclusions

In conclusion, both quantity and quality of food late at night are associated with SVD. Regarding the quantity of food, late-night overeating was positively associated with the existence of carotid plaques and non-systematic late-night food consumption was associated with higher DBP levels. In terms of food quality of systematic late-night food consumers, a dietary pattern consisting of increased red meat consumption, wine, refined grains and reduced consumption of whole grains was positively associated with AIx, therefore with arterial stiffening. As eating late at night concerns many people due to the modern way of life, further investigation is necessary in order to identify dietary behaviors and/or food choices late at night which will be possibly associated with improved vascular health and consequently will enable CVD prevention.

## Figures and Tables

**Table 1 nutrients-14-00470-t001:** Descriptive and clinical characteristics of study population.

	Non-Systematic Late-Night Food Consumers (*n* = 178)	Systematic Late-Night Food Consumers (*n* = 723)	*p*-Value	Systematic Late-Night Eaters (Consumption of <40% of dTEI) (*n* = 572)	Systematic Late-Night Overeaters (Consumption of >40% of dTEI) (*n* = 151)	*p*-Value
**Age (years)**	53.36 ± 13.64	52.16 ± 13.85	0.300	53.05 ± 13.92	48.79 ± 13.10	**0.001**
**Sex (%)**						
Male	48.9	44.3	0.268	39.9	60.9	**0.000**
Female	51.1	55.7		60.1	39.1	
**BMI (Kg/m^2^)**	28.12 ± 5.56	27.93 ± 5.52	0.681	27.83 ± 5.34	28.31 ± 6.15	0.344
**Smoking (%)**						
Never smokers	42.1	39.1	0.626	40.9	32.5	0.104
Former smokers	32.6	36.4		34.6	43.0	
Current smokers	25.3	24.5		24.5	24.5	
**Biochemical biomarkers**						
Fasting blood glucose (mg/dL)	96.62 ± 18.89	99.37 ± 32.45	0.306	99.07 ± 32.70	100.53 ± 31.57	0.670
Total cholesterol (mg/dL)	194.10 ± 37.10	196.15 ± 36.83	0.534	195.32 ± 35.19	199.37 ± 42.62	0.296
LDL-C (mg/dL)	116.53 ± 28.96	118.69 ± 32.15	0.451	117.04 ± 31.34	124.95 ± 34.53	**0.019**
HDL-C (mg/dL)	56.65 ± 21.11	56.18 ± 15.82	0.760	57.34 ± 15.68	51.76 ± 15.66	**0.001**
TG (mg/dL)	110.99 ± 56.38	107.71 ± 66.59	0.570	105.99 ± 56.31	114.35 ± 96.58	0.377
**Diabetes mellitus (%)**						
Type 1 diabetes mellitus	3.4	9.0	**0.040**	78.4	11.3	0.351
Type 2 diabetes mellitus	11.2	11.8		12.4	9.3	
**Hypertension (%)**						
Yes	58.9	48.6	**0.022**	51.2	40.2	**0.037**
**Dyslipidemia (%)**						
Yes	39.3	36.2	0.444	36.2	36.4	0.957
**Subclinical vascular biomarkers**						
Peripheral SBP (mmHg)	127.58 ± 15.66	124.68 ± 15.84	**0.029**	125.12 ± 15.98	123.03 ± 15.24	0.150
Central SBP (mmHg)	117.23 ± 15.27	114.188 ± 16.59	**0.027**	114.49 ± 16.98	113.06 ± 15.03	0.346
DBP (mmHg)	77.88 ± 9.77	75.50 ± 8.97	**0.002**	75.42 ± 8.87	75.82 ± 9.37	0.631
PWV (m/s)	8.39 ± 1.78	8.27 ± 2.00	0.491	8.36 ± 2.07	7.94 ± 1.69	**0.023**
AIx@75 (%)	26.76 ± 11.85	25.78 ± 13.16	0.367	26.61 ± 12.96	22.63 ± 13.46	**0.001**
right IMT (mm)	0.68 ± 0.15	0.68 ± 0.15	0.789	0.68 ± 0.15	0.67 ± 0.15	0.252
left IMT (mm)	0.73 ± 0.17	0.71 ± 0.16	0.182	0.72 ± 0.17	0.70 ± 0.16	0.253
**Existence of plaques (%)**						
Yes	57.9	48.1	**0.020**	47.7	49.7	0.671
**Existence of carotid plaques (%)**						
Yes	42.7	36.4	0.119	35.7	39.1	0.439
Existence of femoral plaques (%)						
Yes	43.3	35.1	**0.044**	34.8	36.4	0.708

dTEI: total energy intake, BMI: body mass index, LDL-C: low-density lipoprotein cholesterol, HDL-C: high-density lipoprotein cholesterol, TG: triglycerides, SBP: systolic blood pressure, DBP: diastolic blood pressure, PWV: pulse wave velocity, AIx: augmentation index, IMT: intima media thickness. Regarding numbers, bold indicates statistically significant results.

**Table 2 nutrients-14-00470-t002:** Dietary intake between non-systematic late-night food consumption, systematic late-night food consumption, late-night food consumption of <40% of dTEI and late-night overeating.

	Non-Systematic Late-Night Food Consumers (*n* = 178)	Systematic Late-Night Food Consumers (*n* = 723)	*p*-Value	Systematic Late-Night Eaters (*n* = 572)	Systematic Late-Night Overeaters (*n* = 151)	*p*-Value
Total energy consumption after 19:00 (Kcal)	415.75 ± 340.63	523.00 ± 375.74	**0.001**	405.31 ± 227.45	968.84 ± 478.90	**0.000**
Energy from CHO after 19:00 (%)	41.77 ± 9.89	42.33 ± 9.69	0.490	42.71 ± 9.36	40.91 ± 10.77	**0.043**
Energy from PRO after 19:00 (%)	16.51 ± 4.96	17.30 ± 4.96	0.058	17.35 ± 5.06	17.08 ± 4.59	0.549
Energy from FAT after 19:00 (%)	40.68 ± 10.36	39.64 ± 8.78	0.174	39.78 ± 8.74	39.12 ± 8.95	0.414
dTEI (Kcal)	1767.79 ± 742.20	1694.95 ± 608.26	0.172	1644.39 ± 558.02	1886.45 ± 740.84	**0.000**
**Food groups**						
Low fat dairy products (250 mL milk or soy milk, 1 cup of yogurt)	0.39 ± 0.62	0.37 ± 0.55	0.712	0.25 ± 0.40	0.20 ± 0.42	**0.001**
Full fat dairy products (250 mL milk, 1 cup of yogurt)	0.18 ± 0.36	0.19 ± 0.40	0.695	0.20 ± 0.42	0.15 ± 0.33	0.146
Vegetables (e.g., 1 medium tomato or pepper, ½ cup of cabbage or peas or lettuce or spinach)	2.21 ± 1.87	2.44 ± 1.88	0.139	2.44 ± 1.77	2.45 ± 2.25	0.933
Legumes cooked and drained (½ cup)	0.41 ± 0.83	0.26 ± 0.59	**0.006**	0.27 ± 0.60	0.21 ± 0.51	0.244
Fruits (e.g., 1 medium apple or peach or orange or pear, 1 cup of strawberries or melon) and fresh juice (½ cup)	1.62 ± 1.89	1.46 ± 1.45	0.228	1.53 ± 1.45	1.20 ± 1.46	**0.014**
Processed juices (½ cup)	0.06 ± 0.24	0.08 ± 0.34	0.512	0.08 ± 0.35	0.08 ± 0.32	0.950
Refined grains (e.g., 30 g of white bread, ½ cup rice or pasta, 2 rusks)	2.93 ±2.40	2.98 ± 2.35	0.810	2.80 ± 2.26	3.66 ± 2.57	**0.000**
Whole grains (e.g., 30 g of whole wheat bread, ½ cup brown or wild rice or whole wheat pasta, 2 rusks)	1.00 ± 1.45	1.07 ± 1.39	0.559	1.08 ± 1.40	1.01 ± 1.31	0.563
Danishes (1 medium slice)	0.13 ± 0.28	0.11 ± 0.29	0.545	0.11 ± 0.28	0.13 ± 0.32	0.462
Low fat cheese (30 g)	0.13 ± 0.34	0.23 ± 0.51	**0.012**	0.21 ± 0.45	0.31 ± 0.70	**0.037**
Full fat cheese (30 g or 1 slice)	0.78 ± 0.97	0.99 ± 1.15	**0.027**	0.96 ± 1.15	1.10 ± 1.15	0.185
Dressings, full fat or light (1 tbs)	0.12 ± 0.38	0.12 ± 0.40	0.994	0.11 ± 0.36	0.19 ± 0.50	**0.023**
Light soft drinks (250 mL)	0.03 ± 0.16	0.07 ± 0.29	0.057	0.07 ± 0.28	0.09 ± 0.36	0.337
Soft drinks with sugar (250 mL)	0.08 ± 0.26	0.07 ± 0.25	0.676	0.06 ± 0.23	0.11 ± 0.32	**0.026**
Low fat sweets (e.g., 1 ice cream 0%, 1 cup of fruit gel)	0.06 ± 0.32	0.07 ± 0.28	0.881	0.06 ± 0.25	0.09 ± 0.42	0.334
Sweets rich in sugar and saturated fat (e.g., 1 slice of cake, 1 ice cream)	0.65 ± 1.01	0.69 ± 0.94	0.639	0.71 ± 0.93	0.62 ± 0.98	0.344
Sugar (1 tsp)	0.94 ± 1.32	0.78 ± 1.27	0.137	0.76 ± 1.27	0.84 ± 1.27	0.462
Sugar substitutes, honey, jam (1 tsp)	0.37 ± 0.93	0.62 ± 1.10	**0.005**	0.65 ± 1.13	0.53 ± 0.97	0.238
Snacks (1 cup of chips, crackers or popcorn)	0.05 ± 0.22	0.04 ± 0.24	0.787	0.04 ± 0.23	0.07 ± 0.26	0.131
High alcohol drinks (e.g., 30 mL of gin, vodka, whiskey, rum)	0.18 ± 0.74	0.14 ± 0.65	0.487	0.13 ± 0.69	0.17 ± 0.50	0.477

dTEI: total energy intake, CHO: carbohydrates, PRO: proteins, tbs: tablespoon, tsp: teaspoon. Regarding numbers, bold indicates statistically significant results.

**Table 3 nutrients-14-00470-t003:** Associations between subclinical vascular biomarkers and A. systematic late-night food consumption or B. late-night overeating.

	A. Systematic vs. Non-Systematic Consumption of Food after 19:00 h			B. Late-Night Food Consumption of >40% of dTEI vs. Late-Night Eating		
	**Model 1**	**Model 2**	**Model 3**	**Model 1**	**Model 2**	**Model 3**
	**B** **(95% CI)**	**B** **(95% CI)**	**B** **(95% CI)**	**B** **(95% CI)**	**B** **(95% CI)**	**B** **(95% CI)**
Peripheral SBP	−2.20 (−4.61, 0.21)	−1.22 (−3.27, 0.83)	−1.15 (−3.20, 0.90)	−1.23 (−3.92, 1.47)	−1.66 (−3.96, 0.64)	−1.83 (−4.14, 0.48)
Central SBP	−2.29 (−4.68, 0.12)	−1.57 (−3.73, 0.59)	−1.52 (−3.69, 0.64)	0.81 (−1.89, 3.51)	0.40 (−2.04, 2.83)	0.32 (−2.13, 2.77)
DBP	**−2.03** **(−3.47, −0.59)**	**−1.45** **(−2.76, −0.13)**	**−1.44** **(−2.76, −0.12)**	−0.06 (−1.64, 1.52)	−0.25 (−1.70, 1.20)	−0.25 (−1.71, 1.20)
AIx@75	−0.83 (−2.48, 0.81)	−0.65 (−2.28, 0.98)	−0.75 (−2.37, 0.88)	0.19 (−1.64, 2.02)	−0.04 (−1.85, 1.78)	0.19 (−1.63, 2.01)
PWV	0.08 (−0.18, 0.35)	0.09 (−0.15, 0.33)	0.09 (−0.15, 0.33)	−0.17 (−0.47, 0.13)	−0.16 (−0.44, 0.11)	−0.16 (−0.43, 0.12)
right IMT	0.00 (−0.02, 0.02)	0.00 (−0.02, 0.02)	0.00 (−0.02, 0.02)	0.01 (−0.01, 0.03)	0.01 (−0.01, 0.03)	0.01 (−0.01, 0,03)
left IMT	−0.01 (−0.03, 0.01)	−0.01 (−0.03, 0.01)	−0.01 (−0.03, 0.01)	0.01 (−0.01, 0.04)	0.01 (−0.02, 0.03)	0.01 (−0.02/0.03)
	**Exp(B) (95%CI)**	**Exp(B) (95%CI)**	**Exp(B) (95%CI)**	**Exp(B) (95%CI)**	**Exp(B) (95%CI)**	**Exp(B) (95%CI)**
Existence of total plaques	0.72 (0.48, 1.07)	0.71 (0.46, 1.09)	0.68 (0.45, 1.05)	1.40 (0.90, 2.17)	1.27 (0.79, 2.04)	3.14 (0.83, 2.16)
Existence of carotid plaques	0.82 (0.56, 1.20)	0.84 (0.57, 1.24)	0.81 (0.54, 1.19)	**1.69** **(1.09/2.61)**	**1.61** **(1.02, 2.53)**	**1.70** **(1.07, 2.68)**
Existence of femoral plaques	0.78 (0.53, 1.15)	0.74 (0.48, 1.13)	0.73 (0.47, 1.12)	1.28 (0.82, 1.99)	1.09 (0.67, 1.78)	1.11 (0.68, 1.80)

dTEI: total energy intake, SBP: systolic blood pressure, DBP: diastolic blood pressure, PWV: pulse wave velocity, AIx: augmentation index, IMT: intima media thickness. Model 1: adjusted for age and sex. Model 2: adjusted for age, sex, hypertension, diabetes mellitus, dyslipidemia, smoking and BMI. Model 3: adjusted for age, sex, hypertension, diabetes mellitus, dyslipidemia, smoking, BMI and dTEI. Regarding numbers, bold indicates statistically significant results.

**Table 4 nutrients-14-00470-t004:** Identification of late-night eating dietary patterns (among systematic late-night food consumers).

Food Groups	Dietary Pattern 1	Dietary Pattern 2	Dietary Pattern 3
Consumption of olive oil	0.770		
Consumption of vegetables	0.740		
Consumption of fish	0.553		
Consumption of cold cuts		0.686	
Consumption of full-fat cheese		0.676	
Consumption of fruits and juices		−0.338	
Consumption of sweets rich in sugar and saturated fat		0.306	
Consumption of red meat			0.658
Consumption of refined grains			0.558
Consumption of whole wheat grains			−0.530
Consumption of wine			0.407
**Explained variance %**	15.661	13.484	12.425

Variables with the highest factor loading (>|0.3|) within the component.

**Table 5 nutrients-14-00470-t005:** Multiple linear and logistic regression analysis between subclinical vascular biomarkers and late-night eating DPs.

	Dietary Pattern 1			Dietary Pattern 2			Dietary Pattern 3		
	**Model 1** **B (95% CI)** ***p*-Value**	**Model 2** **B (95% CI)** ***p*-Value**	**Model 3** **B (95% CI)** ***p*-Value**	**Model 1** **B (95% CI)** ***p*-Value**	**Model 2** **B (95% CI)** ***p*-Value**	**Model 3** **B (95% CI)** ***p*-Value**	**Model 1** **B (95% CI)** ***p*-Value**	**Model 2** **B (95% CI)** ***p*-Value**	**Model 3** **B (95% CI)** ***p*-Value**
**Peripheral SBP**	−0.55 (−1.62, 0.53) 0.320	−0.23 (−1.16, 0.69) 0.618	−0.34 (−1.27, 0.60) 0.480	0.15 (−0.94, 1.23) 0.790	0.40 (−0.53, 1.32) 0.400	0.23 (−0.75, 1.21) 0.645	0.04 (−1.06, 1.15) 0.938	0.31 (−0.63, 1.25) 0.512	0.20 (−0.76, 1.16) 0.684
**Central SBP**	−0.42 (−1.50, 0.66) 0.447	−0.23 (−1.21, 0.74) 0.641	−0.29 (−1.27, 0.70) 0.570	0.08 (−1.00, 1.17) 0.879	0.32 (−0.66, 1.30) 0.523	0.25 (−0.79, 1.29) 0.638	0.26 (−0.84, 1.36) 0.645	0.51 (−0.48, 1.50) 0.313	0.47 (−0.55, 1.48) 0.365
**DBP**	−0.12 (−0.75, 0.51) 0.706	0.04 (−0.54, 0.62) 0.888	0.04 (−0.54, 0.63) 0.886	0.21 (−0.42, 0.85) 0.513	0.31 (−0.27, 0.89) 0.294	0.35 (−0.27, 0.96) 0.267	0.03 (−0.62, 0.67) 0.939	0.12 (−0.47, 0.71) 0.690	0.13 (−0.48, 0.73) 0.683
**AIx@75**	−0.70 (−1.43, 0.03) 0.059	−0.71 (−1.43, 0.02) 0.056	−0.59 (−1.32, 0.15) 0.117	−0.28 (−1.02,0.46) 0.458	−0.21 (−0.94, 0.52) 0.570	0.07 (−0.70, 0.85) 0.850	0.60 (−0.15, 1.34) 0.117	0.63 (−0.11, 1.37) 0.094	**0.84 (0.09, 1.59)** **0.028**
**PWV**	−0.04 (−0.16, 0.09) 0.585	−0.03 (−0.14, 0.09) 0.621	−0.03 (−0.14, 0.09) 0.646	−0.02 (−0.14,0.10) 0.778	0.01 (−0.10, 0.11) 0.926	0.01 (−0.10, 0.13) 0.845	−0.03 (−0.15,0.09) 0.633	0.03 (−0.09, 0.14) 0.638	0.03 (−0.08, 0.15) 0.588
**right ΙΜΤ**	0.00 (−0.01, 0.01) 0.707	0.00 (−0.01, 0.01) 0.619	0.00 (−0.01, 0.01) 0.721	−0.01 (−0.01,0.00) 0.274	0.00 (−0.01, 0.01) 0.535	0.00 (−0.01, 0.01) 0.735	0.00 (−0.01, 0.01) 0.354	0.01 (0.00, 0.02) 0.146	0.01 (0.00, 0.02) 0.093
**left ΙΜΤ**	0.00 (−0.01, 0.01) 0.752	0.00 (−0.01, 0.01) 0.858	0.00 (−0.01, 0.01) 0.933	−0.01 (−0.02,0.00) 0.087	−0.01 (−0.02, 0.00) 0.245	−0.01 (−0.02, 0.00) 0.150	0.00 (−0.01, 0.01) 0.960	0.00 (−0.01, 0.01) 0.616	0.00 (−0.01, 0.01) 0.702
	Exp(B) (95% CI) *p*-value	Exp(B) (95% CI) *p*-value	Exp(B) (95% CI) *p*-value	Exp(B) (95% CI) *p*-value	Exp(B) (95% CI) *p*-value	Exp(B) (95% CI) *p*-value	Exp(B) (95% CI) *p*-value	Exp(B) (95% CI) *p*-value	Exp(B) (95% CI) *p*-value
**Existence of total plaques**	1.00 (0.83, 1.19) 0.965	0.99 (0.82/1.20) 0.906	1.04 (0.85, 1.27) 0.697	0.94 (0.79, 1.13) 0.523	0.99 (0.82, 1.20) 0.919	1.08 (0.88, 1.32) 0.459	1.13 (0.95, 1.36) 0.170	1.11 (0.91, 1.34) 0.298	1.18 (0.96, 1.44) 0.116
**Existence of carotid plaques**	0.87 (0.73, 1.04) 0.128	0.87 (0.72, 1.04) 0.125	0.90 (0.75, 1.09) 0.298	0.92 (0.76, 1.11) 0.380	0.96 (0.79, 1.17) 0.682	1.04 (0.85, 1.28) 0.712	1.13 (0.94, 1.35) 0.196	1.12 (0.92, 1.34) 0.256	1.18 (0.97, 1.43) 0.103
**Existence of femoral plaques**	1.12 (0.94, 1.34) 0.220	1.14 (0.93, 1.39) 0.207	1.17 (0.95, 1.43) 0.143	0.96 (0.80, 1.16) 0.677	1.01 (0.81, 1.24) 0.964	1.04 (0.83, 1.30) 0.756	1.12 (0.93, 1.35) 0.233	1.08 (0.88, 1.32) 0.450	1.10 (0.90, 1.36) 0.347

DP: dietary pattern, SBP: systolic blood pressure, DBP: diastolic blood pressure, PWV: pulse wave velocity, AIx: augmentation index, IMT: intima media thickness. Model 1: adjusted for age and sex. Model 2: adjusted for age, sex, hypertension, diabetes mellitus, dyslipidemia, smoking and BMI. Model 3: adjusted for age, sex, hypertension, diabetes mellitus, dyslipidemia, smoking, BMI and dTEI. Regarding numbers, bold indicates statistically significant results.

## Data Availability

The data presented in this study are available on request from the corresponding author. The data are not publicly available due to protection of privacy of the participants’ medical data, that can be found in their files.

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
