# Peer review of "Late-Night Overeating or Low-Quality Food Choices Late at Night Are Associated with Subclinical Vascular Damage in Patients at Increased Cardiovascular Risk"

_nutrients, 2022, doi:10.3390/nu14030470_

Round 1
Reviewer 1 Report
Thank you for the opportunity to review the paper “Late-night overeating or low-quality food choices late at night are associated with subclinical vascular damage in patients at increased cardiovascular risk.” by Eirini D. Basdeki et al.
This study moves from the evidence that the consumption of a large amount of daily total energy intake during the night, is associated with cardiovascular risk factors such as hypertension, dyslipidemia, obesity, glycemic profile disorders. Nevertheless, limited data exist regarding the association with subclinical vascular damage, that precedes the onset of established cardiovascular diseases, such as atheromatic plaques, increased diastolic blood pressure, intima-media thickness, aortic stiffness, and so on. The authors conducted a cross-sectional study in a large sample of adults that underwent anthropometric, dietary, and vascular assessment, and demonstrated that both quantity and quality of food late at night are associated with the risk factors above mentioned, especially with carotid plaques and higher diastolic blood pressure levels; moreover, increased consumption of certain food - such as red meat, wine, refined grains - is associated with arterial stiffening.
Researchers conducted good quality work, the writing is clear and the study method appropriate. I consider this paper of striking interest to readers. As long as that late-night overeating concerns more and more people due to modern lifestyle changes, it would be interesting to have other similar studies to identify dietary behaviors and types of food that should be avoided or, conversely, that could be associated with better vascular health and lower rates of cardiovascular diseases.
Minor remarks:
- Abstract: “Aix”: please, report in extenso the acronym
- Introduction, line 81 (was consisted): remove "was"
Reviewer 2 Report
I read with interest the manuscript entitled ‘ Late-night overeating or low-quality food choices late at night 2 are associated with subclinical vascular damage in patients at 3 increased cardiovascular risk’. The present well-designed and presented study evaluates the effect of LNO on cardiovascular risk factors. Only, a few points should be considered.
- The authors should explain the acronym AIx in the abstract.
- The authors mention that LNO might have potential cardiovascular effects. I would not reach this conclusion based on the noticed DBP difference.
- Is seems quite interesting that the majority of the study participants were LNO. The authors should underline this.
- For the performed analyses i would prefer the term ‘after adjusting for’ rather than ‘independently of sex, age, etc’
- Are there any available data on concomitant medication? If no, this should be stated as a limitation. Similarly, the lack of data on subjects’ physical activity is a major limitation of the present study.
